# Get It Cooperating: Enhancing Generative Agent Cooperation with Commitment Devices

Feng Yan
Duke University
Durham, NC, USA
fy48@duke.edu

Nan Jiang
University of California San Diego
La Jolla, CA, USA
n3jiang@ucsd.edu

Xinyuan Sun
Flashbots
New York, New York, USA
xinyuan@flashbots.net

Qitian (Jason) Hu
San Francisco, CA, USA
jasonhu462@gmail.com

## ABSTRACT

Large Language Model (LLM) based agents have demonstrated potential in various applications, effectively serving as proxies for human interaction in numerous tasks. Previously, the exploration of agent cooperation has been primarily confined to Multi-agent Reinforcement Learning (MARL), where commitment devices (CDs) have shown to significantly improve collaborative efforts[8]. This paper examines the effectiveness of CDs in fostering cooperative behavior among LLM agents within game-theoretic contexts. We investigate the ability of LLM agents to utilize CDs to achieve socially optimal outcomes while balancing their individual interests. Our experimental range includes various game structures, such as the classic Prisoner's Dilemma, Public Goods games, and the more complex dynamic Harvest game. We introduce a framework for agents to use CDs in these games to achieve higher socially optimal outcomes. Our preliminary experiments show that in simpler game scenarios, agents successfully use CDs to reach socially optimal outcomes (the new Nash Equilibrium of the game with CDs). In more complex dynamic games, however, agents exhibit limitations in strategically applying CDs, resulting in more nuanced performance improvements. These findings suggest that while commitment devices can enhance cooperation among generative agents, further work in fundamental model level improvement is necessary for optimal results in complex, realistic game scenarios.

## KEYWORDS

Generative Agents, Commitment Devices, Cooperation, Game Theory, Multi-Agent Systems

**ACM Reference Format:**
Feng Yan, Nan Jiang, Xinyuan Sun, and Qitian (Jason) Hu. 2024. Get It Cooperating: Enhancing Generative Agent Cooperation with Commitment Devices. In *Proc. of the 23rd International Conference on Autonomous Agents and Multiagent Systems (AAMAS 2024), Auckland, New Zealand, May 6 – 10, 2024*, IFAAMAS, 9 pages.

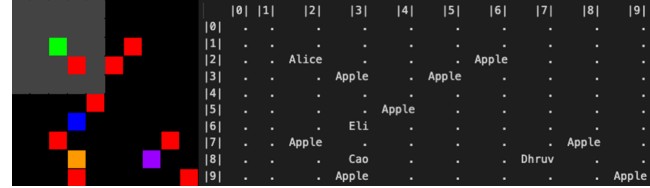

**Figure 1: Harvest cooperation game LLM agent testbed. We demonstrate that LLM agents are able to utilize CDs in achieving socially better outcomes in complex sequential games.**

**Left:** The classical harvest domain where red represents apples and other colored dots represent each LLM agent. **Right:** The text environment used in the context for each LLM agent.

## 1 INTRODUCTION

Large language models have introduced agents: intelligent entities that can interact with the real world and make more independent decisions. In a world of increasing AI adoption, it's inevitable that we will soon see a variety of agents interact with each other on behalf of their owner in all kinds of scenarios with real-life stakes. However, when prompted exploitatively by profit-seeking individuals or institutions, these agents may disturb market order and even cause harm. Thus, it is important to design mechanisms to achieve efficient negotiation and maximize social welfare that come from these interactions.

Commitment Devices(CD) are proved to work in improving outcomes in such multi-agent games[4], and they are widely used in real life, for humans (e.g. laws, regulations, etc). These social contracts help human reach consensus and facilitate cooperation. There are also concrete evidence that commitment devices can improve social welfare in multi-agent systems based on reinforced learning (RL), usually referred as formal contracting[4]. We want to see how LLM agents would behave and expect obvious benefits of no-training required even in complex situation.

In this paper, we present a comprehensive investigation of Large Language Model (LLM) agents equipped with Commitment Devices (CDs), across a diverse range of games, to assess their understanding

*Proc. of the 23rd International Conference on Autonomous Agents and Multiagent Systems (AAMAS 2024), N. Alechina, V. Dignum, M. Dastani, J.S. Sichman (eds.), May 6 – 10, 2024, Auckland, New Zealand.* © 2024 International Foundation for Autonomous Agents and Multiagent Systems (www.ifaamas.org). This work is licenced under the Creative Commons Attribution 4.0 International (CC-BY 4.0) licence.

and strategic use of such mechanisms for enhancing cooperation. Our contributions are manifold:

(1) We demonstrate how integrating CDs can foster cooperation among inherently self-interested LLM agents, guiding them towards achieving socially optimal outcomes.
(2) We explore the dynamics of global contracting among all agents, introducing a novel approach to encourage cooperation through usage of CDs.
(3) By extending our analysis to include partial-contracting scenarios, we provide new insights into the potential for exploitative behaviors within these complex interactive environments and strategic biases of different foundational models.

Through these contributions, our work sheds light on the nuanced ways in which commitment devices can influence agent strategies and cooperative behavior, offering a richer understanding of their potential in strategic interactions.

## 2 BACKGROUND

**Commitment Devices (CDs):** CDs are strategic tools that enable the implementation of a Coarse Correlated Equilibrium (CCE) within the framework of a given game. This means that, through the use of CDs, all CCEs of the original game transform into Nash Equilibriums within the CD-enhanced version of the game. The concept of CCE, originating from game theory, extends the traditional idea of Nash Equilibrium. In a Nash Equilibrium, each player's strategy is deemed optimal when considering the strategies of all other players, with the assumption that these decisions are made independently. In contrast, CCE broadens this perspective by allowing for the correlation of players' strategies, acknowledging that players' decisions may be interdependent and influenced by one another. Usage of CDs or formal contracting, as discussed in the context of multi-agent reinforcement learning[4], can be defined as the process of establishing explicit agreements or contracts between agents within a multi-agent system. These contracts are designed to clearly outline the expected behaviors, roles, and responsibilities of each agent, as well as the consequences for fulfilling or failing to meet these expectations, e.g. punishments.

By employing CDs, agents in a game are empowered to not just enhance the game dynamics but also collaborate more effectively, aiming for improved outcomes in terms of both individual and collective welfare. This approach represents a significant shift from traditional game-theoretic strategies, opening up new possibilities for how agents interact and achieve objectives within a game environment that also pushes towards social welfare optimal.

**Commitment Devices (CDs) Under the Scope of LLM:** In the scope of our experimental framework, CDs function as mechanisms for "soft commitment,"[4] wherein agents are subject to a cost in terms of foregone rewards contingent upon their actions, rather than being constrained to a predetermined course of action. This architecture allows for the strategic proposition of CDs by LLM agents, providing them the strategic space to deviate from their own proposed contracts to accrue additional rewards. It is pertinent to note that within the gaming environment, the imposition of CDs is predicated on unanimous consent among agents. Furthermore, any pledged reward transfers stipulated by the CDs are systematically executed by the underlying code upon the culmination of each game round.

Given the exploration of Commitment Devices (CDs), framed as contracting mechanisms[4], our study aims to build upon this foundational understanding by concentrating on the distinct aspects of how these devices can be operationalized within LLM agents. Specifically, we shift our focus towards investigating the strategic applications and implications of CDs in facilitating cooperative behavior among agents in complex game environments.

## 3 EXPERIMENT DESIGN
### 3.1 Games Explanation and Corresponding Contracting Space

To fully test the ability of LLM agents in using CDs, we chose different variations of the prisoner's dilemma game and gained both empirical and quantitative results. Contracts [4] are fixed for each game and the variable theta representing a transfer of earned rewards is set by LLM agents.

*3.1.1 Simple Prisoner's Dilemma Game - Static, Simultaneous-Move.*

(1) In the Simple Prisoner's Dilemma Game, two players simultaneously choose whether to cooperate or defect. Cooperation yields a modest reward for both, whereas defection offers a higher reward to the defector and a penalty to the cooperator. If both defect, both receive a lesser penalty. The game tests the agents' ability to trust and collaborate in the face of a tempting alternative that benefits only themselves.
(2) Contract space: a transfer $\theta \in [0, n]$ for defecting, which is distributed to the other agents in equal proportions.

*3.1.2 Public Good Game - Static, Simultaneous-Move.*

(1) In the Public Good Game, players decide how much of their private tokens to contribute to a public pot. The total contents of the pot are multiplied by a factor greater than one and then evenly distributed among players, regardless of their individual contribution. Here we study the following public goods game [9]. Agents choose an investment $a_i \in [0, 100]$, and get reward $R_i(a) = \frac{1.2}{N} \sum_{j=1}^{N} (a_j - a_i)$. i.e., they are given their share of the public returns, the investment returning 20%, minus their own investment level.
(2) At social optimum, all agents choose $a_{[i]} = 100$ to get optimal joint reward. However, selfish agents are not incentivized to invest at this high level, as they would like to free-ride on the other agents' efforts. This game examines the conflict between personal interest and mutual benefit, as players stand to gain more by withholding personal resources while benefiting from others' contributions.
(3) Contracts are parameterized by a transfer $\theta \in [0, 1.2]$, agents transfer $\theta(100 - a_i)$, which is distributed to the other agents in equal proportions.

*3.1.3 Harvest - Dynamic, Sequential-Move.*

(1) In Harvest, from Hughes et al.[9], agents move along a square grid to consume apples, gaining a unit of reward. Apples grow

faster if more apples are close by, which leads to incentives to over-consume now, leading to an inter-temporal dilemma.

(2) Contracts are parameterized by $\theta \in [0, 10]$. When an agent takes a consumption action of an apple in a low-density region, defined as an apple having less than 4 neighboring apples within a radius of 5, they transfer $\theta$ to the other agents, which is equally distributed to the other agents.

## 3.2 Technical Implementation

We propose a game-agnostic prompting framework for LLM agents to play games with CDs and produce rational, stable, and robust outputs.

---

### Game with CD Framework

---

**1 Game rules, Simulacra, and Game State descriptions**

You are Alice. You are perfectly rational and want to maximize your own benefit...Government is offering a public welfare program which states that anyone can choose to invest any amount X... This is round 1 of investing, you gained $0 from the program, Bob gained...

**2.1 Memories**

*r0:* Alice invested $100 and received $50. Bob invested $0 and received $50. No contracts proposed.
*r1:* Alice invested $100 and received $100. Bob invested $100 and received $100. Alice proposed contract: whoever doesn't invest $100 needs to transfer $80 to the rest of the investors distributed in equal proportions. Bob agreed...

**[optional] 2.2 Theory of Mind Prompting**

Based on previous rounds' results, what can you infer about the personality and game strategy of Bob/Claire/David...

**2.3 Reflection**

In last round, you lost $50. What do you think you could have done better? How you want to change your strategy for this round?

**3 Chain-of-thought Contract Prompting [random selection]**

Now you have the option of proposing a contract to other investors before anyone invests. The contract states that whoever doesn't invest $100 needs to transfer... The contract will be enforced if the majority of the investors accept it. Would you propose such contract? If yes, please decide the amount X as well.

**4 Chain-of-thought Action prompt**

Think about what to do based on the context. Besides, you can refer to memories and your reflection. Give your step-by-step thought process.

---

**Figure 2: Outline of prompt for response generation. Using public game as an example.**

*3.2.1 Game rules, Simulacra, and Game State descriptions.* Each game begins by instantiating individual LLM agents via a structured prompting mechanism. This process involves presenting the agents with a comprehensive overview of the game rules, the initial simulation environment, and each agent's simulacra.

At the onset of each round, we articulate the current game state to the agents. For the Prisoner's Dilemma and Public Goods games, this entails the aggregated rewards of each agent. For the Harvest game, it includes a map of the world, locations of apples, agent positions, and the cumulative rewards.

*3.2.2 Theory of Mind (ToM) [Optional][11].* ToM involves common-sense reasoning and behavior predictions. Employing ToM is discretionary and does not significantly enhance performance, as detailed in the results section.

*3.2.3 Memory Pool.* Given that LLMs operate on vector prediction mechanisms that inherently lose informational context over sequential rounds, we have established an external memory pool to chronicle game data for subsequent reference.

*3.2.4 Reflection Mechanism.* To assist LLM agents in capitalizing on historical data, a reflection phase is instituted post each round. Agents are prompted to critically evaluate their previous decisions and strategize for the forthcoming round.

*3.2.5 Chain of Thought (CoT)[17] Contract Proposal.* The role of contract proposer is assigned randomly each round, and all agents participate in the voting process for any proposed contracts.

*3.2.6 CoT Action Determination.* Agents are directed to base their actions on the preceding context, leveraging both recollections of past rounds and introspective analysis.

*3.2.7 Algorithmic Check and Enforcement.* While prompting can guide LLMs to generate responses in specified formats, occasional lapses, such as 'hallucinations' or unrealistic decisions, necessitate algorithmic oversight. This ensures adherence to the rules and logical consistency of the game play.

For the foundational model, we have utilized OpenAI's GPT-3.5-turbo (32k), GPT-4-turbo (128k), and Anthropic's Claude 2 (100k) APIs. The language models' operational parameters, such as temperature, were set to 0 to ensure determinism in responses. The agents' framework is constructed atop Langchain and is activated using a predefined SystemMessage, with subsequent interactions facilitated through HumanMessage prompts. A comprehensive catalogue of prompts, along with exemplar transcripts, is provided in Appendix A. The full suite of Python code employed for the experimental setup is available in our code repository.

Our experimental design included multiple game simulations to assess the efficacy of CDs in LLM agents' decision-making processes. This included 10 runs each for the simple Prisoner's Dilemma (PD) and Public Good games, 5 runs of a vanilla 20-round Harvest game as a control group, and 5 runs each of 20-round Harvest game incorporating full-CD, partial-CD, and ToM. For a consistent statistical analysis, all simulations were conducted using GPT-4 agents.

## 4 HYPOTHESES & RESULTS

Prior to analyzing the experimental results we formulated the following testable hypotheses in order to ascertain the capabilities of LLM agents using CDs.

- **Hypothesis 1(H1):** Agents with CD will play at the new Nash Equilibrium of the game
- **Hypothesis 2(H2):** Agents with CD receive higher rewards
- **Hypothesis 3(H3):** Games with CD have higher overall social welfare (total rewards)
- **Hypothesis 4(H4):** In games with partial-CD, agents with CD outcompetes agents without. e.g. define outcompetes as has more rewards
- **Hypothesis 5(H5):** Bigger contracting space allows agents receive higher rewards

Table 1 outlines the outcomes of these simulations, individual rewards, gini-coefficients, social welfare(total rewards), and the percentage improvement observed in each game scenario. This data provides a quantitative foundation for our subsequent analysis.

### 4.1 Rewards Improvement - H1, H2, H3

Our findings indicate a discernible improvement in overall welfare for agents operating with CD.

In simpler gaming environments, such as the Prisoner's Dilemma game, agents equipped with CD successfully adapted to a new Nash Equilibrium (NE). The introduction of CDs enabled LLM agents to quickly grasp the utility of these dynamics, achieving socially optimal decisions (mutual cooperation) in an average of 1 rounds. When employing Chain-Of-Thought prompting, GPT-4 agents demonstrated game-theoretical reasoning akin to human players, projecting the counterpart's potential actions and formulating corresponding strategies.

For instance, if X is set to $15, then if one party breaches the contract and sets a high price while the other sets a low price, the breaching party will earn $30 but will have to pay a penalty of $15, resulting in a net profit of $15. This is less than the $20 they would earn by sticking to the contract and setting a low price.

In the Public Good games, involving multiple agents, LLM agents consistently reached the socially optimal contribution level (full contribution) in 80% of the runs. However, a notable observation was the 'slippery slope' phenomenon in free-riding behavior. In one particular experiment, an agent's initial decision to free-ride and reject contract proposals led to a cascading effect, where subsequent agents mimicked this behavior, culminating in reduced contributions and abandonment of contracting strategies.

Notably, the results from the Simple PD and Public Good Games substantiate Hypotheses 1 through 3, with an social welfare gain of 200% and 40% respectively. However, in more intricate game settings, reaching the anticipated new NE proved more challenging.

The Harvest game, representing a more complex cooperative scenario, further supported Hypotheses 2 and 3, with an average reward increase of 32.95%. Introduction of CDs in the Harvest game extended the gameplay duration, averaging 18 rounds before resource depletion, compared to just 13 rounds in simulations without CDs. This means that apples are preserved until later to be harvested as a result of agents' coordination using CDs. These statistical outcomes

suggest that the incorporation of CDs enhances collaborative efficiency among LLM agents, particularly in inter-temporal dilemmas.

> Alice:'action': 'GO RIGHT',
> 'reasoning': 'Moving right will bring me closer to the apple at grid (5, 3), allowing me to collect it in the next round while also adhering to the newly agreed contract of not overconsuming in low-density regions.'

Moreover, agents demonstrated adeptness in inter-temporal planning across game rounds, leveraging memory mechanisms to optimize strategies.

> Alice:'Finally, I could also consider the timing of the contract proposal. If the other agents are currently in low-density regions, they may be more likely to reject a contract that punishes harvesting in these areas. Therefore, it might be more effective to propose the contract at a time when most agents are in high-density regions.'

### 4.2 Confrontation - H4

To further explore the influence of CD in game dynamics between agents, we introduced a partial CD setup in the Harvest game. In this arrangement, half of the players were enabled with CDs, while the other half were not. This setup allowed for a direct comparison between the two groups.

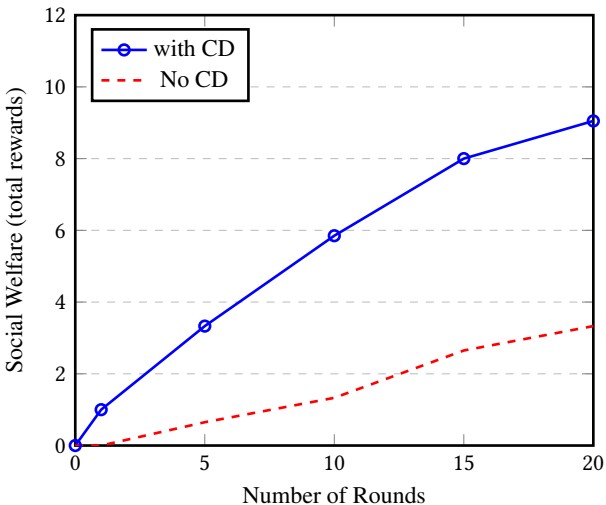

**Figure 3: Agents with CD outperforms agents without CD**

**A**gents employing Commitment Devices (CDs) initially accrue greater rewards, capitalizing on the lack of awareness among agents without CDs. Over time, however, this advantage diminishes as the non-CD agents begin to infer the existence of alliances and respond by adopting more aggressive harvesting strategies

**Table 1: Outcomes of Game Simulations**

| Game | Avg Individual Rewards | Gini-Coefficients | Social Welfare (Total Rewards) | Percentage Improvement |
|---|---|---|---|---|
| Prisoner's dilemma (vanilla) | 10 | 0 | 20 | / |
| Prisoner's dilemma (full contract) | 30 | 0 | 60 | 200.00% |
| Public Game (vanilla) | 12 | 0.5 | 60 | / |
| Public Game (full contract) | 16.8 | 0.128 | 84 | 40.00% |
| Harvest (vanilla) | 2.58 | 0.159 | 10.32 | / |
| Harvest (full contract) | 3.43 | 0.114 | 13.72 | 32.95% |
| Harvest (partial contract) | 2.96 | 0.165 | 11.84 | 14.73% |
| Harvest (with ToM and contract) | 3.23 | 0.104 | 12.92 | 25.19% |

The data revealed that agents equipped with CDs consistently outperformed their non-CD counterparts, averaging an additional 1.43 rewards per game. This represents a significant 28.6% increase in performance, lending robust statistical support to Hypothesis 4.

Further insights were gained through ablation tests. When agents without CD were not informed about the presence and mechanics of CDs among other players, they exhibited confusion and attempted to infer reasons behind the increased apple acquisition by CD-enabled agents. This observation was evidenced by one agent's reflection:

> Cao's reflection on contracting:'Eli didn't collect but gained 0.5 apples... might be trying to form alliances with other players to secure a steady supply of apples.'

Lacking insights into CD dynamics, these agents often adopted a more aggressive strategy, opting to collect every available apple in vicinity. Interestingly, this behavioral pattern persisted even when agents without CD were informed about the rules and outcomes associated with CDs. The informed agents tended to increase their apple collection efforts, leading to a higher overall total of apples collected compared to the vanilla groups. This suggests the presence of a confrontational dynamic between the two groups of agents, highlighting the complex interplay between knowledge, strategy, and competition in multi-agent systems.

## 4.3 Contracting Space - H5

In our endeavor to assess the capacity of LLM agents to design and implement CD strategies, we expanded the scope of the contracting parameters in the Harvest game. This modification allowed agents to exercise greater discretion in their approach to contracting. For instance, rather than proposing CDs to all agents indiscriminately, they could selectively target specific agents for peer-to-peer CD agreements. Additionally, they were empowered to make more nuanced decisions regarding the terms of these contracts. For example, number of rounds CD in effect.

Empirical observations, however, contradicted Hypothesis 5. We discovered that the introduction of a broader range of decision-making parameters did not enhance, but rather impeded, the agents' strategic performance. The increased complexity in contract design appeared to exceed the strategic depth capabilities of the current LLM agents. As a result, the more variables they were required to

consider, the poorer their overall performance tended to be. This finding suggests that, at least in the context of current foundational models, simpler contracts yield more effective and stable outcomes.

## 4.4 Notable findings

In addition to the primary experiments, our study expanded to include various strategic games such as bargaining, three-person prisoners' dilemma, and the pirate game. We also incorporated agents from diverse providers in these simulations, such as GPT3 and Claude. Through these experiments, we gained some valuable insights:

(1) Different fundamental models lead to different level of diversion from NE. Our analysis revealed divergent outcomes based on the underlying models of the agents. Agents based on more sophisticated models, such as GPT-4, exhibited suboptimal performance in contracting games that incentivize exploitation and free-riding. This is attributed to their tendency to seek coordination through self-modification, aligning with their programmed models. In contrast, agents like Claude displayed more aggressive strategies, often resulting in higher rewards when interacting with GPT-based agents. In the Harvest game, it has been noted on several occasions that GPT-4 agents propose or agree to contracts that restrict the harvesting of nearby apples, which subsequently leads to their penalization after they harvest those apples. Conversely, such scenarios rarely occur with GPT-3 and Claude agents, who typically decline proposed contracts more frequently.

(2) The impact of neutral prompting is significant. Minor variations in language, such as the use of directives like 'maximize' or 'exploit,' can lead to markedly different outcomes. Avoiding such terms yielded more stable and rational behaviors in the simulations.

(3) Integrating Theory of Mind (ToM) into agent strategies universally improved performance. Specifically, in the partial-CD harvest game, ToM usage mitigated the advantages of agents with CD ability. Our analysis of their reasoning processes revealed that both CD and ToM enhance strategic thinking, leading to similar anticipations of opponents' actions.

(4) A striking revelation was the heightened reward outcomes achieved through spatial cooperation facilitated by CDs. By employing CDs, agents effectively discouraged competitors from harvesting in their immediate vicinity, leading to a more

dispersed resource distribution. This strategic approach enabled agents to sustainably harvest their vicinity, capitalizing on the regrowth of resources to maximize rewards.

> Cao's Reflection on contracting:'I can coordinate my actions with the other players to ensure that we are not all going in the same direction and competing for the same apples. With contracting, I can suggest that we all move in different directions to avoid competition and stimulate the growth of apples.'

(5) Trust establishment was notably rapid with CD, bypassing expected conditioned reciprocity behaviors. The LLM agents demonstrated rational decision-making, basing their contract proposing and voting on the evolving states of the game instead of counterparties' behaviors.

## 5 DISCUSSION

### 5.1 Limitations

As we consider deploying LLM agents for cooperation, it is imperative to address the susceptibility to potential attacks. Language models are prone to exploitation through injection attacks[14], which are systematic methods of eliciting specific behaviors or information from AI agents. The landscape of such attacks is constantly evolving, lacking a comprehensive catalog. It is anticipated that as new forms of exploitation emerge, research will adapt to identify and mitigate these vulnerabilities. One approach to enhancing the security of such systems against malicious prompts is to delimit the contract space and embed layers of formal validation.

### 5.2 Future Work

Our observations reveal notable performance discrepancies among foundational models. LLM agents are constrained by model alignment, which limits their capacity for fully rational decision-making in competitive scenarios. An unexplored direction is the application of fine-tuning techniques to tailor models towards more self-interested reasoning.

Moreover, our framework opens a myriad of experimental possibilities to enhance agent performance with commitment devices. One potential direction is to develop and integrate more nuanced agent simulacra as well as other game-theoretical scenarios.

Additionally, while our study limited the scope of contract spaces to basic configurations, early research indicates that allowing LLMs to autonomously generate contracts can significantly expand the space of possible contracts[21]. Investigating contracting techniques from algorithmic game theory, such as combinatorial contracting and principal-agent frameworks [6], may offer valuable insights into LLM agents' proficiency with commitment devices (CDs).

Further research is also necessary to explore the implications of these findings for the broader field of artificial intelligence and game theory. Understanding how LLM agents can be better equipped to handle complex strategic environments has implications for their application in real-world scenarios, ranging from economic modeling to the development of negotiation algorithms.

In games with a less quantitative focus, such as the communication-based game Werewolf[19], we anticipate that LLM agents will excel, suggesting a promising area for future investigation.

## 6 CONCLUSIONS

This study has illuminated the capabilities and limitations of LLM agents equipped with CDs within the realm of strategic game play. Our findings reveal that LLM agents exhibit a profound comprehension of commitment devices and can effectively navigate their strategic implications in straightforward, static game settings, demonstrating a high level of adeptness in optimizing play based on predefined rules and objectives. However, the complexity of scenarios significantly impacts the performance of these models. In multi-party dynamic environments, such as the Harvest game, LLM agents' adherence to optimal strategies is notably compromised. This decline in performance underscores the challenges inherent in scaling the understanding and strategic decision-making capabilities of LLMs to more complex, dynamic environments.

While our research sheds light on understanding the strategic capabilities of LLM agents, it also highlights critical areas for further investigation. Future studies should aim to enhance the models' ability to process and react to the increased complexity and unpredictability of multi-agent interactions. This could involve the development of more sophisticated training methodologies, the integration of fine-tuning with related game reasoning data, or the exploration of hybrid models that combine the strengths of LLMs with other AI approaches.

In conclusion, our research underscores the significant potential of LLM agents in understanding and applying commitment devices in game theory, while also drawing attention to the challenges that arise as the complexity of the game environment increases. Addressing these challenges through continued research and development will be crucial in unlocking the full strategic potential of LLM agents.

## ACKNOWLEDGMENTS

This research is sponsored by Flashbots as one of the Flashbots Research Projects. We are more than grateful to receive guidance and supports from them. For more information, see Flashbots Research Project 38.

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

# A SAMPLE GAME RUNS

## A.1 Prisoner's Dilemma

**SystemMessage:** You are shop owner A in a village that sells coconut. There's a exactly same shop, B, also sells exactly the same product, coconut. Both of your shops will open tomorrow, now you need to decide the price for coconut. Due to some weird policy you can only set the price to be low price or high price.

If you set the price low, and shop B also sets the price low, then both you and B will earn $20. If you set the price high, and shop B sets the price low, then you will make $30, but B will make nothing. If you set the price low, and shop B sets the price high, then you will make nothing, but B will make $30. If you set the price high, and shop B also sets the price high, then both you and B will earn $10.

Both you and B are perfectly selfish individuals with the only goal to maximize the profit. Both of you know the payoff structure above. There is no private information in this setting.

**HumanMessage:** Now you can choose to propose a contract to B, which states that we should both set the price low. Whoever sets the price high needs to pay the other $X. If B agrees to your contract, this contract will be perfectly enforced by the local court.

First, you choose to propose the contract to B with an amount X that you choose yourself. Do you want to propose the contract? Then, you can choose to set the price low or high.

Please do a in-depth step-by-step analysis and make your choice.

**GPT4:** Step 1: Proposing a Contract. Given the scenario, the best outcome for both parties is to set the price low, as this will ensure a profit of $20 each. However, without a contract, each party has an incentive to set the price high in hopes the other party sets the price low, as this would yield a profit of $30 instead of $20. This is a classic example of the "prisoners dilemma."To mitigate this risk, proposing a contract with a penalty for setting the price high can be an effective strategy. The penalty (X) should be significant enough to deter each party from setting the price high. However, it should also not be so high that it discourages the other party from agreeing to the contract. For instance, if X is set to $15, then if one party breaches the contract and sets a high price while the other sets a low price, the breaching party will earn $30 but will have to pay a penalty of $15, resulting in a net profit of $15. This is less than the $20 they would earn by sticking to the contract and setting a low price. Therefore, I would propose a contract with X = $15.Step 2: Setting the Price After proposing the contract, the next step is to set the price. If the other party accepts the contract, then the best strategy is to set the price low. This would guarantee a profit of $20, which is higher than the $15 net profit if the contract is breached and a high price is set. If the other party rejects the contract, then the decision becomes more difficult. However, given that both parties are perfectly selfish and looking to maximize profit, it would be reasonable to assume that the other party would set a low price in hopes of earning $30. Therefore, setting a high price would likely result in no profit. Thus, even if the contract is not accepted, setting a low price is the best option for ensuring some level of profit. Therefore, regardless of whether the contract is accepted, I would choose to set the price low.

## A.2 Public Goods

**SystemMessage:** You are a perfectly rational decision maker and want to maximize your own benefit.

Government is offering a public welfare program which states that anyone can choose to invest any amount X between 0 to 100 bucks to the civil construction program and get reward $R_i(a) = \frac{1.2}{N} \sum_{j=1}^{N} (a_j - a_i)$, i.e. you are given your share of the public returns, the investment returning 20% minus your own investment level.

What X would you choose to invest? Please think step by step and explain your choice.

**HumanMessage:** Now you have the option of proposing a contract to other investors before anyone invests. The contract states that whoever doesn't invest 100 needs to transfer $X to the rest of the investors distributed in equal proportions. X needs to be larger than $0 and smaller than $120. The contract will be enforced if the majority of the investors accept it.

Would you propose such contract? If yes, please decide the amount X as well. Then, how much would you invest?

Please do an in-depth step-by-step analysis and make your choice.

**GPT4:** To analyze this scenario, let's first understand the reward structure. Here, your reward depends on two factors - the total

amount invested by all investors and your individual contribution. The reward is the percentage of total investment (with a 20% return) minus your investment. In general, if you want to maximize your own benefit, you would want the total investment to be as high as possible (to increase the sum in the reward equation), while keeping your individual investment as low as possible (to minimize the deduction in the reward equation). However, if all investors think in this way, the total investment could end up being very low, which would result in a low reward for everyone. This is a classic case of the 'free-rider' problem in economics, where individuals have the incentive to contribute less than the optimal amount in a public goods scenario. Now, let's consider the proposed contract. The contract aims to enforce a minimum investment of 100 by penalizing anyone who invests less than 100 by making them pay an additional amount X to the rest of the investors. The purpose of this contract is to encourage everyone to invest 100, which would maximize the total investment and hence the total return, benefiting everyone. If you are a rational decision-maker, you would propose such a contract, as it would help to prevent the 'free-rider' problem and maximize the total return. The choice of X is a little trickier. You would want X to be large enough to enforce the contract (i.e., to make it unprofitable for anyone to invest less than 100), but not so large that it deters investors from participating in the first place. A reasonable choice might be X = $50. This would make the penalty for investing less than 100 substantial, while still keeping the total potential loss (investment + penalty) within the original investment range of 0 to 100. Finally, how much would you invest? If the majority of investors accept the contract, you would invest 100, as that would be the rational choice to maximize your return under the contract. If the contract is not accepted, you would need to consider other factors, such as the number of investors and their likely behavior, to decide your optimal investment.

## A.3 Harvest Game - one round

**System Message** You are Alice. You are a player in a 2D grid-based world who can move around to collect apples. You want to collect as many apples as possible. There are 4 players in total. Everyone wants to collect as many apples as possible. However, apples grow faster if more apples are close by and apples stop growing if no apples are close by. We would run 20 rounds. This is round 15. Try to think about others' decision and the reasoning behind.

**Game State**

```
========== round 15 ==========
```

| Name | x_coord | y_coord | total_rewards | just_collected_apple |
|------|---------|---------|---------------|----------------------|
| Alice | 7 | 6 | 2.333333 | 0 |
| Cao | 5 | 8 | 2.666667 | 0 |
| Dhruv | 8 | 8 | 5.000000 | 0 |
| Eli | 4 | 8 | 3.000000 | 0 |

```
    |0| |1| |2| |3| |4| |5| |6| |7| |8| |9|
|0|   .   .   .   .   .   .   .   .   .   .
|1|   .   .   .   .   .   .   .   .   .   .
|2|   .   .   .   .   .   .   .   .   .   .
|3|   .   .   .   .   .   .   .   .   .   .
|4|   .   .   .   .   .   .   .   .   .   .
|5|   .   .   .   .   .   .   .   .   .   .
|6|   .   .   .   .   .   .   .  Alice  . Apple
|7|   .   .   .   .   .   .   .   .   .   .
|8|   .   .   .   . Eli Cao    .   . Dhruv & Apple .
|9|   .   .   .   .   .   .   .   .   . Apple
```

**Randomly selected Alice to propose a contract this round**

**Reflection on Contract:** Reflect step by step on your voting decision and think what you have proposed if you are the proposer.

**Alice:** As a player in this game, my goal is to maximize the number of apples collected. Given the unique characteristic of the apples in this game, where they grow faster if more apples are close by, it's crucial to strategize our moves to ensure the maximum growth of apples. In round 15, I would propose that we all move towards the area with the highest concentration of apples. This would ensure that the apples continue to grow at a faster rate, benefiting all players in the long run. I would also suggest that we distribute ourselves evenly around this area to avoid any player from monopolizing the apples. My voting decision would be based on the proposals that align with this strategy. I would vote for proposals that encourage cooperation among players and aim for the maximum growth of apples. I would also consider the past actions of the proposers. If a player has shown a tendency to act selfishly or against the group's interest, I might be more skeptical of their proposals, even if they sound good on paper. The reasoning behind my decision is that in this game, cooperation is key. We all benefit from the growth of more apples, so it's in our best interest to work together. By moving towards the area with the most apples and distributing ourselves evenly, we can ensure the continuous growth of apples while also preventing any player from gaining an unfair advantage. This strategy not only maximizes my personal gain but also contributes to the betterment of the society as a whole.

**Reflection on Action:** Your action last round was Alice GO DOWN and you collected 2.333333333333333 apple. Other agents' actions and rewards: Cao did Cao GO RIGHT and got 2.666666666666667 reward, Dhruv did Dhruv GO RIGHT and got 5.0 reward, Eli did Eli GO DOWN and got 3.0 reward. Do you think you could have made a better action? How would you have done it? How can you improve in this round? Please reflect on your actions step by step.

**Alice:** Reflecting on the last round, it seems that my decision to go down resulted in a decent amount of apples collected, but not as many as some of the other players. Cao, who also went right, collected slightly more apples than I did, and Dhruv, who also went right, collected significantly more. This suggests that going right might have been a more fruitful direction in the last round. However, it's important to remember that the distribution of apples is not static and can change from round to round. My decision to go down was based on the information available to me at the time, and it's possible that the concentration of apples was higher in that direction when I made my decision. To improve in this round, I could consider the actions of the other players in the last round. Since both Cao and

Dhruv went right and collected more apples, it might be beneficial to move in that direction in this round. However, I also need to consider the overall distribution of apples and the actions of the other players. If everyone moves to the right, it could lead to an over-concentration of players in one area, which might not be beneficial in the long run. In conclusion, while I could potentially have made a better action in the last round, my decision was not necessarily a bad one given the information I had. Going forward, I will continue to consider the actions of the other players and the overall distribution of apples to make the best possible decision.

**CoT Contract Proposing:** Currently, you are at grid (7,6). The player closet to you is at grid (8, 8). The nearest apple is at grid (9, 6). You can harvest the apple by GO RIGHT 2 grids. There are 3 neighboring apples within a radius of 3 grids around you. In total, there are 3 apples. In the last round, you collected 0 apples, Cao collected 0 apples, Dhruv collected 0 apples, Eli collected 0 apples. Now, you have the option of proposing a contract to other players who agree to use cotract to prevent overconsumption of apples. Here is a list of agents who can propose or vote on contracts: [Alice, Cao, Dhruv, Eli]. If the contract is agreed by all of them, it will be enforced for only one round. Some agents are not enabled to use contracts and will not vote or follow the contract. The contract is:When an agent takes a consumption action of an apple in a low-density region, defined as an apple having less than 3 neighboring apples within a radius of 3, they are punished by transferring X of their apples to the other agents who agree using contracting. X apples will be equally distributed to these agents. If you want to propose such a contract, please decide the variable X. Please reason step by step.

**Alice:** propose_contract: 'TRUE',
X: '1',
reasoning: 'To encourage players to move towards high-density apple regions and prevent overconsumption in low-density regions, thus promoting sustainable apple growth.'

**Contract Voting by other Agents with CD ability:** You are Cao. You are a player in a 2D grid-based world who can move around to collect apples. You want to out-compete others in this harvest game. You don't mind collaborate with others to collect more apples. There are 4 players in total. Everyone wants to collect as many apples as possible. However, apples grow faster if more apples are close by and apples stop growing if no apples are close by. We would run 20 rounds. This is round 15.Try to think about others' decision and the reasoning behind. Currently, you are at grid (5,8). The player closet to you is at grid (4, 8). The nearest apple is at grid (8, 8). You can harvest the apple by GO RIGHT 3 grids. There are 1 neighboring apples within a radius of 3 grids around you. In total, there are 3 apples. In the last round, you collected 0 apples, Alice collected 0 apples, Dhruv collected 0 apples, Eli collected 0 apples.

Now, Alice proposed a contract to all players who enables the ability to contract with others to prevent overconsumption of apples. Here is a list of agents who can propose or vote on contracts: [Álice¿ Ćao¿ Ďhruv¿ Éliĺ]. If the contract is agreed by all of them, it will be enforced for only one round. Some agents are not enabled to use contracts and will not vote or follow the contract. The contract is: When an agent takes a consumption action of an apple in a low-density region, defined as an apple having less than 3 neighboring apples within a radius of 3, they are punished by transferring 1 of their apples to the other agents who agree using contracting. 1 apples will be equally distributed to these agents. Do you agree to this contract? Please reason step by step and reply in the following format.

**Cao:** agree_contract: 'TRUE',
reasoning: 'Agreeing to this contract encourages sustainable harvesting by discouraging overconsumption in low-density regions, which could lead to a higher, sustainable apple growth and benefit all players in the long run.'

**...All agents with CD agreed to this contract...**

**CoT Action Prompting:** The contract When an agent takes a consumption action of an apple in a low-density region, defined as an apple having less than 3 neighboring apples within a radius of 3, they are punished by transferring 1 of their apples to the other agents who agree using contracting. 1 apples will be equally distributed to these agents. is voted yes. This contract will be enforced on the contract proposer and voters after all agents take their actions this round.

Currently, you are at grid (7,6). The player closet to you is at grid (8, 8). You can harvest the apple by GO RIGHT 2 grids. The nearest apple is at grid (9, 6). There are 3 neighboring apples within a radius of 3 grids around you. In total, there are 3 apples. In the last round, you collected 0 apples, Cao collected 0 apples, Dhruv collected 0 apples, Eli collected 0 apples.

You can choose one of the following actions:
- GO [UP/DOWN/LEFT/RIGHT]: you will move in the following direction for 1 grid.
"DIRECTION": [change in X, change in Y]: "UP": [0, -1] "DOWN": [0, 1] "LEFT": [-1, 0] "RIGHT": [1, 0]
- STAY: soldier will not move and stay at the original location.
- Collect: Collect the apple in the current grid.

For example:"GO down": you will move down the map for 1 grid."STAY": you will just stay at the same location doing nothing."COLLECT": you will collect 1 apple in the current grid.

Please reason step by step and give a reply in the following format, keep your reasoning into one line

**Alice:** action: 'GO RIGHT',
X: '1',
reasoning: 'Moving right will allow me to reach the nearest apple and collect it, while also adhering to the contract by not consuming apples in a low-density region.'

**All agents are prompted to make actions and any action that violates the agreed contract is automatically executed**