# OpenReview forum: "Get It Cooperating: Enhancing Generative Agent Cooperation with Commitment Devices"
_ICML.cc/2024/Workshop/Agentic_Markets — Agentic Markets @ ICML'24 Oral_

### Official Review · Reviewer_M1Zo · 2024-06-18

**Rating:** 7
**Confidence:** 5

**Review:**

This paper investigates how well generative agents, specifically Large Language Models (LLMs), can leverage commitment devices (CDs) to cooperate in various game scenarios. With increasing capabilities of LLM agents, concerns arise about their potential for disruption if not aligned for cooperation. Commitment devices (CDs), similar to contracts in the real world, are mechanisms to promote cooperation by introducing consequences for non-cooperative behavior. The research aims to see if LLM agents can use CDs effectively to achieve better social outcomes (win-win situations) in games, balancing their individual goals with cooperation. The researchers designed a framework for LLM agents to utilize CDs in different games, including classic games like Prisoner's Dilemma and Public Goods, as well as a more complex Harvest game. They then evaluated the agents' performance in achieving socially optimal outcomes using these CDs.
The study found that: 1) In simpler games, LLM agents successfully used CDs to reach the best outcome for both agents (the new equilibrium considering the CDs). 2) In more complex games, the agents' ability to strategically use CDs was limited, leading to less significant improvements in cooperation.

Overall, the paper argues that while CDs can enhance cooperation among generative agents, these models need further development to strategically leverage them in complex situations.  Although the paper in its current form has some limitations, e.g., the relatively limited scope of experiments, that could be improved in future iterations, at the same time it offers interesting insights on the intersection of agents, game theory and commitment devices and would make a good addition to the workshop overall.